# Short-term vital parameter forecasting in the intensive care unit: A benchmark study leveraging data from patients after cardiothoracic surgery

Nils Hinrichs[1,2], Tobias Roeschl[1,2,3], Pia Lanmueller[1,4], Felix Balzer[2], Carsten Eickhoff[5], Benjamin O'Brien[4,6,7], Volkmar Falk[1,3,4,8], Alexander Meyer[1,3,4,9]*

1 Department of Cardiothoracic and Vascular Surgery, Deutsches Herzzentrum der Charité (DHZC), Berlin, Germany, 2 Institute of Medical Informatics, Charité - Universitätsmedizin Berlin, Berlin, Germany, 3 Berlin Institute of Health, Charité – Universitätsmedizin Berlin, Berlin, Germany, 4 German Centre for Cardiovascular Research (DZHK), Partner site Berlin, Berlin, Germany, 5 Institute for Bioinformatics and Medical Informatics, Eberhard-Karls-Universität Tübingen, Tübingen, Germany, 6 Department of Cardiac Anesthesiology and Intensive Care Medicine, Deutsches Herzzentrum der Charité (DHZC), Berlin, Germany, 7 Department of Perioperative Medicine, St Bartholomew's Hospital and Barts Heart Centre, London, United Kingdom, 8 Department of Health Sciences and Technology, Translational Cardiovascular Technologies, Eidgenössische Technische Hochschule Zürich, Zürich, Switzerland, 9 Berlin Institute for the Foundations of Learning and Data (BIFOLD), Technical University of Berlin, Berlin, Germany

* alexander.meyer@dhzc-charite.de

## Abstract

Patients in an Intensive Care Unit (ICU) are closely and continuously monitored, and many machine learning (ML) solutions have been proposed to predict specific outcomes like death, bleeding, or organ failure. Forecasting of vital parameters is a more general approach to ML-based patient monitoring, but the literature on its feasibility and robust benchmarks of achievable accuracy are scarce. We implemented five univariate statistical models (the naïve model, the Theta method, exponential smoothing, the autoregressive integrated moving average model, and an autoregressive single-layer neural network), two univariate neural networks (N-BEATS and N-HiTS), and two multivariate neural networks designed for sequential data (a recurrent neural network with gated recurrent unit, GRU, and a Transformer network) to produce forecasts for six vital parameters recorded at five-minute intervals during intensive care monitoring. Vital parameters were the diastolic, systolic, and mean arterial blood pressure, central venous pressure, peripheral oxygen saturation (measured by non-invasive pulse oximetry) and heart rate, and forecasts were made for 5 through 120 minutes into the future. Patients used in this study recovered from cardiothoracic surgery in an ICU. The patient cohort used for model development (n = 22,348) and internal testing (n = 2,483) originated from a heart center in Germany, while a patient subset from the eICU collaborative research database, an American multicenter ICU cohort, was used for external testing (n = 7,477). The GRU was the predominant method in this study. Uni- and multivariate neural network models proved to be superior to univariate statistical models across vital parameters and forecast horizons, and their advantage steadily became more pronounced for increasing forecast horizons. With this study, we established

**Data Availability Statement:** The anonymized German single-center data source used in this study can be obtained upon reasonable request by

contacting the Clinical Data Science working group of Deutsches Herzzentrum der Charité (office-meyer@dhzc-charite.de). Access to the eICU collaborative research database (Pollard T, Johnson A, Raffa J, Celi L A, Badawi O, Mark R. eICU Collaborative Research Database [version 2.0]. PhysioNet. 2019. Available from: https://doi.org/10.13026/C2WM1R) can be obtained by researchers upon request and completion of the required training course. For more information, see https://eicu-crd.mit.edu/gettingstarted/access/. The software code developed as part of this study is available at https://github.com/nhinrichsberlin/icu-vital-parameter-forecasting.

**Funding:** This work was supported by the Berlin Institute of the Foundations of Learning and Data (BIFOLD). The funder had no role in study design, data collection and analysis, decision to publish, or preparation of the manuscript.

**Competing interests:** Pia Lanmueller declares relevant financial activities outside the submitted work with Abbott GmbH und Abiomed Inc. Felix Balzer reports funding from Medtronic and grants from the German Federal Ministry of Education and Research, grants from the German Federal Ministry of Health, grants from the Berlin Institute of Health, personal fees from Elsevier Publishing, grants from Hans Böckler Foundation, other from Robert Koch Institute, grants from Einstein Foundation, and grants from Berlin University Alliance, all outside the submitted work. Carsten Eickhoff is co-founder of Codiag AG and x-cardiac GmbH, and reports consulting fees from Abacus Health and speaker fees from Merck, all outside the submitted work. Benjamin O'Brien declares research funding from the British Heart Foundation and the National Institute for Health Science Research and consulting/speaker fees from Abiomed and Edwards. Volkmar Falk declares relevant financial activities outside the submitted work with following commercial entities: Medtronic GmbH; Biotronik SE & Co.; Abbott GmbH & Co. KG; Boston Scientific; Edwards Lifesciences; Berlin Heart; Novartis Pharma GmbH; JOTEC GmbH; Zurich Heart. In relation to: Educational Grants (including travel support); Fees for lectures and speeches; Fees for professional consultation; Research and study funds. Alexander Meyer declares the receipt of consulting and lecturing fees from Medtronic, lecturing fees from Bayer, and consulting fees from Pfizer. Alexander Meyer is founder and managing director of x-cardiac GmbH. The other authors have no competing interests.

an extensive set of benchmarks for forecast performance in the ICU. Our findings suggest that supplying physicians with short-term forecasts of vital parameters in the ICU is feasible, and that multivariate neural networks are most suited for the task due to their ability to learn patterns across thousands of patients.

## Author summary

The current health status of patients in an Intensive Care Unit (ICU) is continuously tracked through multiple vital parameters, and physicians use these markers to immediately detect life-threatening derangements and for treatment decision-making. Knowledge about future vital parameter values could lead to the anticipation and timely initiation of potentially life-saving interventions. We therefore sought to test how reliably vital parameters of patients in an ICU could be forecast. Vital parameters of interest were blood pressure (diastolic, systolic, and mean), central venous pressure, peripheral oxygen saturation, and heart rate. Our study cohort consisted of patients recovering from cardiothoracic surgery in one German and multiple American ICUs. Using patient data from roughly 22,000 ICU admissions, we developed nine forecast models, ranging in complexity from very simple to highly sophisticated and tested their performance on roughly 10,000 additional ICU admissions by making them forecast values for all six vital parameters over the next two hours. We thus generated an extensive collection of benchmarks for forecast accuracy in the ICU for future researchers to compare against. We found that, compared to simple statistical methods, sophisticated techniques capable of learning patterns from thousands of ICU stays are slightly better at forecasting the immediate next value, and much better when it comes to forecasting further ahead into the future.

## Introduction

Postoperative care in an intensive care unit (ICU) is essential for patients undergoing cardiothoracic surgery [1]. Due to the strain on patients' cardiovascular system caused by the procedure and the underlying health condition, complications can arise which require extensive pharmacological support [2], re-admission to surgery, or organ replacement therapy [3]. Therefore, continuous monitoring of patients' vital parameters is crucial to allow timely and well-targeted interventions.

The high density of structured patient data makes the ICU a prime use case for the implementation of machine learning (ML) solutions, and the feasibility of predicting events like intra-hospital mortality [3–5], length of ICU stay [4], post-operative bleeding [3], renal failure [3], delirium [6] or threshold alarms [7] is well-documented (see [8–10] for reviews). However, due to the multitude of potential complications, a limited number of prediction models for specific adverse events might not be able to capture all threats to the patient's recovery, while further adding to the number of parameters in need of evaluation by the medical staff.

Forecasting vital parameter progression curves instead of specific events offers a more generalizable approach to ML-assisted monitoring in the ICU. Displaying forecasts for upcoming vital parameter values could benefit physicians and patients through early indication of improvements or deteriorations, and could aid with risk-stratification, adjustment of medication, or discharge decisions based solely on parameters which are already constantly and routinely monitored.

Despite its potential to enhance patient monitoring, there is currently a lack of research on forecasting vital parameters in the ICU. Available studies are limited in terms of sample size, or external validation, and report only on a limited set of vital parameters and forecast horizons [11,12]. Moreover, to the best of our knowledge, no studies comparing multiple forecasting methodologies of differing complexity to identify the most suitable approaches in the ICU have been published to date.

While the dominance of modern ML techniques over conventional statistical models has been well established for applications such as image recognition or natural language processing, ML has only in recent years made it into the methodological mainstream of time series forecasting [13]. While recent results, e.g. in forecasting competitions, show that well-designed ML models can be employed highly effectively for forecasting, far simpler and computationally cheaper methods like exponential smoothing remain competitive [13,14]. For the task of vital parameter forecasting in the ICU, it therefore remains to be established whether an increase in model complexity yields better forecasting performance.

With this work, we intend to establish robust benchmarks of forecast accuracy in postoperative cardiothoracic intensive care. For six essential and routinely captured vital parameters recorded at five-minute intervals, we report and compare forecast performance 5 to 120 minutes into the future, using forecasting models differing in complexity from univariate statistical methods to multivariate neural networks designed for sequential data. We hypothesize that high-quality forecasting of vital parameters in the intensive care unit is feasible, and that the forecast quality of modern neural network techniques is superior to that of univariate statistical methods due to their ability to perform cross-patient learning and pattern detection using data from thousands of ICU stays at once.

## Materials and methods

### Study populations

We used two data sources in this study: an internal dataset used for model development, hyperparameter optimization and internal testing, and an external dataset exclusively used for model testing.

Model development, hyperparameter optimization and internal testing were performed on an anonymized single-center dataset from a German quaternary cardiovascular center. It includes vital parameters recorded at one-minute intervals during ICU stays of patients following cardiothoracic surgery between October 2012 and August 2022.

For external testing of our proposed methods, we used a sub-set of patients from the eICU Collaborative Research Database, a multi-center database containing intensive care data at five-minute intervals from the United States collected in 2014 and 2015 [15].

Vital parameters of interest were systolic, diastolic, and mean blood pressure, central venous pressure, heart rate and peripheral oxygen saturation (measured via non-invasive pulse oximetry). Other data sources like laboratory values or medication were not part of this study.

For compatibility of the two data sources, we down-sampled vital parameters in the internal single-center cohort to five-minute intervals using the median. To avoid data leakage during down-sampling, we kept the first timepoint ($t_0$) and value untouched, and defined values at later timepoints ($t_0 + 5$ minutes, $t_0 + 10$ minutes . . .) as the median of the current and four preceding observations.

Inclusion criteria were identical for both data sources. We included only adult postoperative patients aged 18 or older, admitted to a cardiothoracic surgery ICU. We required at least one hour of recorded ICU data per patient, and included only the first 24 hours of their first

ever recorded ICU admission. Re-admissions were excluded from the analysis. Thus, each ICU admission corresponds to one unique patient and vice versa.

On the internal single-center cohort, we performed a random 8:1:1 split across patients into a training set, a hyperparameter validation set, and an internal test set.

We obtained permission from the Charité–Universitätsmedizin Berlin ethics committee to retrospectively analyze the single-center data (approval number EA1/202/23) and obtained access to the eICU database.

## Data cleaning

Using pre-defined bounds of plausibility for each vital parameter (Table 1), we removed values exceeding these limits and considered them missing. In the internal dataset, this was done prior to down-sampling. The start of each patient's multivariate vital parameter time-series was defined as the first timepoint with at least one valid non-missing vital parameter. Per vital parameter, we imputed missing values as follows: we used forward-filling for at most three consecutive timepoints where a previous value of the same patient was known. If more than three consecutive timepoints were missing or no previous value was known, we imputed using non-missing values of the other vital parameters through scikit-learn's iterative imputing strategy with linear regression models to predict missing from available vital parameters at a given timepoint. [16]

## Forecast methodology

We implemented nine different forecasting models from three model classes (Table 2): five univariate statistical models, two univariate neural networks (N-BEATS [17] and N-HiTS [18]), and two multivariate neural networks, specifically a recurrent neural network with gated recurrent unit (GRU) [19] and a Transformer architecture specifically designed for time-series tasks [20].

Fig 1 schematically depicts a synthetic example of our forecasting setup for three successive forecasts produced by one method for a single vital parameter. Every five minutes, models produced forecasts for 5, 10, 15, . . ., 120 minutes into the future. Forecasts were always based on the entire observed past up to the current time point. Thus, the length of the forecast window remained fixed at 120 minutes, but the amount of past values on which predictions were based depended on the time point of prediction.

Univariate statistical models were newly trained from scratch at every time point per vital parameter and patient, which allowed models to change with each new observation. In contrast, for uni- and multivariate neural networks, we trained single, immutable models on all patients observed in the training set. For test set predictions, at each time point per patient, these immutable models were presented with the vital parameters up to the point in question, from which forecasts were computed.

**Table 1. Upper/lower thresholds used for data cleaning.** Values exceeding these limits were treated as missing.

|  | Lower limit | Upper limit |
| --- | --- | --- |
| **Systolic blood pressure (mmHg)** | 8 | 350 |
| **Diastolic blood pressure (mmHg)** | 8 | 150 |
| **Mean blood pressure (mmHg)** | 8 | 170 |
| **Central venous pressure (mmHg)** | -15 | 50 |
| **Heart rate (1/min)** | 10 | 330 |
| **Oxygen saturation (%)** | 40 | 100 |

**Table 2. Overview of forecasting models.**

| Model | Class | Training | Implementation | Reference |
|---|---|---|---|---|
| Naive | Univariate statistical | Trained from scratch for every forecast | *forecast*[24], R | [23] |
| Theta | Univariate statistical | Trained from scratch for every forecast | *forecast*, R | [21,26] |
| ETS | Univariate statistical | Trained from scratch for every forecast | *forecast*, R | [22,23] |
| ARIMA | Univariate statistical | Trained from scratch for every forecast | *forecast*, R | [23] |
| AR NNet | Univariate statistical | Trained from scratch for every forecast | *forecast*, R | [23] |
| N-BEATS | Univariate neural network | One model per vital parameter, trained on the training set | *NeuralForecast*[27], Python | [17] |
| N-HiTS | Univariate neural network | One model per vital parameter, trained on the training set | *NeuralForecast*, Python | [18] |
| GRU | Multivariate neural network | One model for all vital parameters, trained on the training set | *PyTorch*[29], Python | [19] |
| Transformer | Multivariate neural network | One model for all vital parameters, trained on the training set | *PyTorch*, Python | [20] |

Per patient and vital parameter, we produced forecasts every five minutes starting with the sixth, and ending with the penultimate timepoint. Thus, univariate statistical models were given at least five past observations to train on, and all forecasts could be evaluated on at least one true value.

As the only manual adjustment of forecasts produced by our models, forecasts exceeding the pre-defined bounds of plausibility (Table 1) were capped at the corresponding limit.

## Univariate statistical models

Due to the non-seasonal and non-cyclic nature of the studied vital parameters at five-minute intervals, we restricted the choice of univariate statistical models to models without explicit components for these types of patterns. We implemented five univariate statistical forecasting models: the naïve method (constant forecast using the most recent observed value, hereafter referred to as Naive), the Theta method [21], exponential smoothing (ETS) [22], the autoregressive integrated moving average (ARIMA) model [23], and an autoregressive single-layer neural network (AR NNet) [23]. All five methods were implemented using R's forecast package [24]. Model specifications were iteratively tested on the validation set to optimize performance in terms of the root mean squared error, but also to avoid any numerical errors during model fitting or at prediction time.

If at least 24 past observations were available for a given vital parameter, we used Box-Cox-transformation [25] on the time series with a transformation parameter between zero and two, using the implementation in the forecast package [24]. Predictions were inverse-transformed to retain the original scale.

For the ETS model, we defined the following potential model specifications and let the forecast package chose the optimal specification based on the corrected Akaike Information Criterion (AICc). We used only non-seasonal ETS models with either additive or multiplicative error type, and either additive, multiplicative or no trend. Further, we allowed the trend to be either damped or non-damped. [22–24]

The potential ARIMA model specifications were as follows: We allowed no seasonal model components or seasonal differencing. Non-seasonal first-order differencing was an option. We restricted the autoregressive order and the moving average order to 1, 2 or 3, each, with their sum not exceeding 4. The optimal model was chosen based on the AICc. [23,24]

The AR NNet was of non-seasonal autoregressive order 6, included no seasonal autoregressive term, a hidden size of 18 and a weight decay of 2. We emphasized recent observations through exponentially rising observation weights. The final forecasts were the result of fitting five models with different initial model weights and averaging their individual results. [23,24]

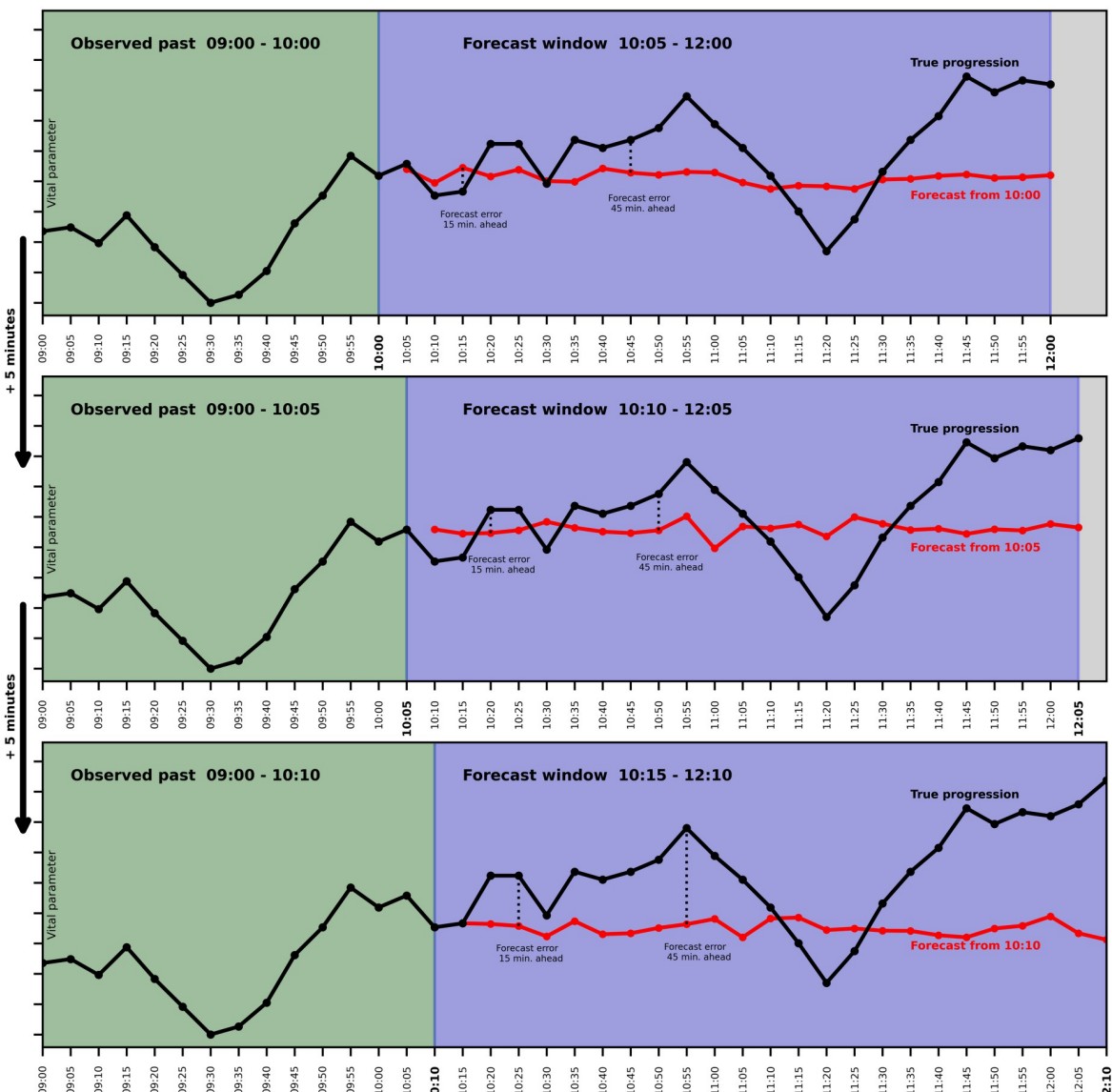

**Fig 1. Schematic illustration of the forecasting setup.** Synthetic example of three successive forecasts produced by the same method for one simulated vital parameter. Layout of the figure inspired by [11].

We implemented the Theta model, a combination of simple exponential smoothing with a linear trend obtained through simple regression [26], using the forecast package, but manually overrode checks for seasonality to force unseasonal models [24].

## Univariate neural network models

N-BEATS (short for Neural Basis Expansion Analysis for interpretable Time Series forecasting) [17] and N-HiTS (Neural Hierarchical Interpolation for Time Series forecasting) [18], two state-of-the-art univariate forecasting models based on multiple stacks of multi-layer perceptrons, were implemented using the Neural Forecast [27] python package. Given a set of potential hyperparameters per model, the package provides automatic selection and fitting of

the best-performing model using a rolling window approach which continually validates model performances.

Per vital parameter, both models were fitted on the training set. Using the mean and standard deviation estimated from the training set, vital parameters were standardized to exhibit mean zero and standard deviation one prior to model fitting, and de-standardized post prediction. Model training lasted 100 epochs in batches of size 128. Per vital parameter and model, 25 hyperparameter configurations were tested using the HyperOpt [28] search algorithm.

For both models, hyperparameter choices were identical for the following three parameters: the learning rate (between $5 \cdot 10^{-5}$ and $5 \cdot 10^{-3}$ in steps of $10^{-5}$), the number of hidden multi-layer perceptron units (32, 64 or 128) and the number of past timepoints used as model input size (12, 24, 48 or 72). Additional hyperparameters were the polynomial degree for trend modeling in the N-BEATS model (choices 2, 3 or 4), and dropout (between 0.0 and 0.4 in steps of 0.01) in the N-HiTS model.

## Multivariate neural network models

We implemented both the GRU and the Transformer using PyTorch [29]. Models with different hyperparameter choices were fitted on the training set, and their performance was evaluated on the validation set. Vital parameter standardization was performed as described above for univariate neural network models. The models were designed to output forecasts for all 24 forecast horizons (5 through 120 minutes) simultaneously.

Hyperparameter choices for the GRU were the number of layers (between 1 and 4), the hidden size per layer (128, 256 or 512), dropout (between 0.0 and 0.4 in steps of 0.01), L2 regularization (between $10^{-6}$ and $5 \cdot 10^{-5}$ in steps of $10^{-6}$), and the learning rate (between $5 \cdot 10^{-5}$ and $5 \cdot 10^{-3}$ in steps of $10^{-5}$).

For the Transformer, we used the time-series architecture proposed by Zerveas et al (2021) [20]. In addition to the hyperparameters used for training the GRU, it required hyperparameter choices for the activation function (rectified linear unit or Gaussian error linear unit), positional encoding (fixed or learnable), model dimension (64, 128, 256 or 512), and the number of transformer heads (8 or 16).

Hyperparameter optimization was implemented using the Ray package [30]. We used the HyperOpt search algorithm [28] with trials set up and ended prematurely in case of poor performance after at least 10 epochs using the Hyperband algorithm [31]. Models were trained for up to 100 epochs with training data presented in batches of 128 patients. 50 different hyperparameter combinations were tried to find the most suitable model.

## Statistical analysis

Forecasts were compared to the known ground truth as displayed in Fig 1 for all forecast horizons from 5 to 120 minutes into the future. For all six vital parameters and all forecast horizons, we computed the root mean squared error (RMSE) to evaluate the model accuracy. If the ground truth was unknown (and the vital parameter subsequently imputed to allow model fitting), the observation was not counted towards the RMSE. We used repeated bootstrapping on the patient level with 250 iterations to compute 95% confidence intervals. As additional error metrics reported in the appendix, we computed the mean absolute error (MAE) and the mean absolute percentage error (MAPE).

To summarize model performances in a single metric, we further computed the MAPE, a scale-less error metric, across all vital parameters and forecast horizons. 95% confidence intervals for MAPE were again computed via bootstrapping.

**Table 3. Description of the study populations.** Continuous variables are reported as Median [Q1, Q3]. Binary variables are reported as n (%). BP = Blood pressure. SpO2 = Peripheral oxygen saturation (measured via non-invasive pulse oximetry).

| | Training data | Validation data | Internal test data | External test data (eICU) |
|---|---|---|---|---|
| No of observations | 3 980 262 | 501 843 | 493 548 | 1 997 745 |
| No of patients | 19 865 | 2 483 | 2 483 | 7 477 |
| Age (years) | 68.00 [58.00, 76.00] | 68.00 [58.00, 75.00] | 68.00 [59.00, 76.00] | 68.00 [59.00, 76.00] |
| Sex (male) | 2 749 272 (69.1) | 343 267 (68.4) | 346 117 (70.1) | 127 5870 (63.9) |
| BP Systolic (mmHg) | 116.00 [101.00, 132.00] | 116.00 [101.00, 131.00] | 116.00 [101.00, 131.00] | 117.93 [106.00, 130.00] |
| BP Diastolic (mmHg) | 56.37 [50.00, 63.00] | 56.63 [50.00, 63.00] | 56.00 [50.00, 63.00] | 56.41 [50.00, 63.00] |
| BP Mean (mmHg) | 75.00 [68.00, 83.00] | 75.00 [68.00, 83.00] | 75.00 [68.00, 83.00] | 75.79 [69.00, 82.00] |
| Central venous pressure (mmHg) | 9.00 [6.00, 12.00] | 9.00 [6.00, 12.00] | 9.00 [6.00, 12.00] | 9.12 [8.00, 10.94] |
| Oxygen saturation (%) | 98.00 [96.00, 100.00] | 98.00 [96.00, 100.00] | 98.00 [96.00, 100.00] | 97.41 [96.00, 99.00] |
| Heart rate (1/min) | 89.00 [78.00, 96.00] | 89.00 [79.00, 96.00] | 89.00 [78.00, 96.00] | 80.00 [71.00, 90.00] |

A code repository containing the relevant software behind this project can be found at https://github.com/nhinrichsberlin/icu-vital-parameter-forecasting.

## Results

### Study populations

We identified 24 831 patients in the internal dataset, out of which 19 865 were randomly allocated to the training set, 2 483 into the validation set, and 2 483 to the internal test set. The relevant cohort from the eICU database used for external testing consisted of 7 477 patients (Table 3).

Patients' ages were similar across datasets, with a median of 68 years. However, the prevalence of male sex was higher in the internal sets (69.1, 68.4, 70.1% in training, validation, test set, respectively) compared with the eICU dataset (63.9%). For systolic, diastolic, and mean blood pressure, as well as peripheral oxygen saturation, there were no major discrepancies in the respective distribution between the populations. The external test set exhibited fewer patients with low central venous pressure (lower quartile 6.0 in the internal cohorts vs. 8.0 in the external test set), and, on average, lower heart rates (median 89.0 in the internal cohorts vs. 80.0 in the external test set). A detailed comparison of vital parameter distributions across cohorts is depicted in S1 Fig.

In the internal dataset, missing vital parameters were scarce, ranging from 1.8% for oxygen saturation to 3.1% for the heart rate. In the eICU cohort, the rate of missing values was considerably higher for systolic, diastolic, and mean blood pressure (~21% each) as well as central venous pressure (52.2%). Details are given in S1 Table.

### Comparison of forecast quality

An example of a single forecast run of four models at a single timepoint is illustrated in Fig 2. The mean absolute percentage error across all vital parameters and forecast horizons is displayed in Figs 3 (internal test set) and 4 (external test set). Root mean squared errors per vital parameter, forecast model and selected forecast horizons are displayed in Figs 5 (internal test set) and 6 (external test set). A complete summary of performances for all vital parameters and forecast horizons using three accuracy metrics (RMSE, MAE and MAPE) can be found in S2 Table. Optimal hyperparameters of the uni- and multivariate neural network models are listed in S3 (Transformer), S4 (GRU), S5 (N-BEATS) and S6 (N-HiTS) Tables.

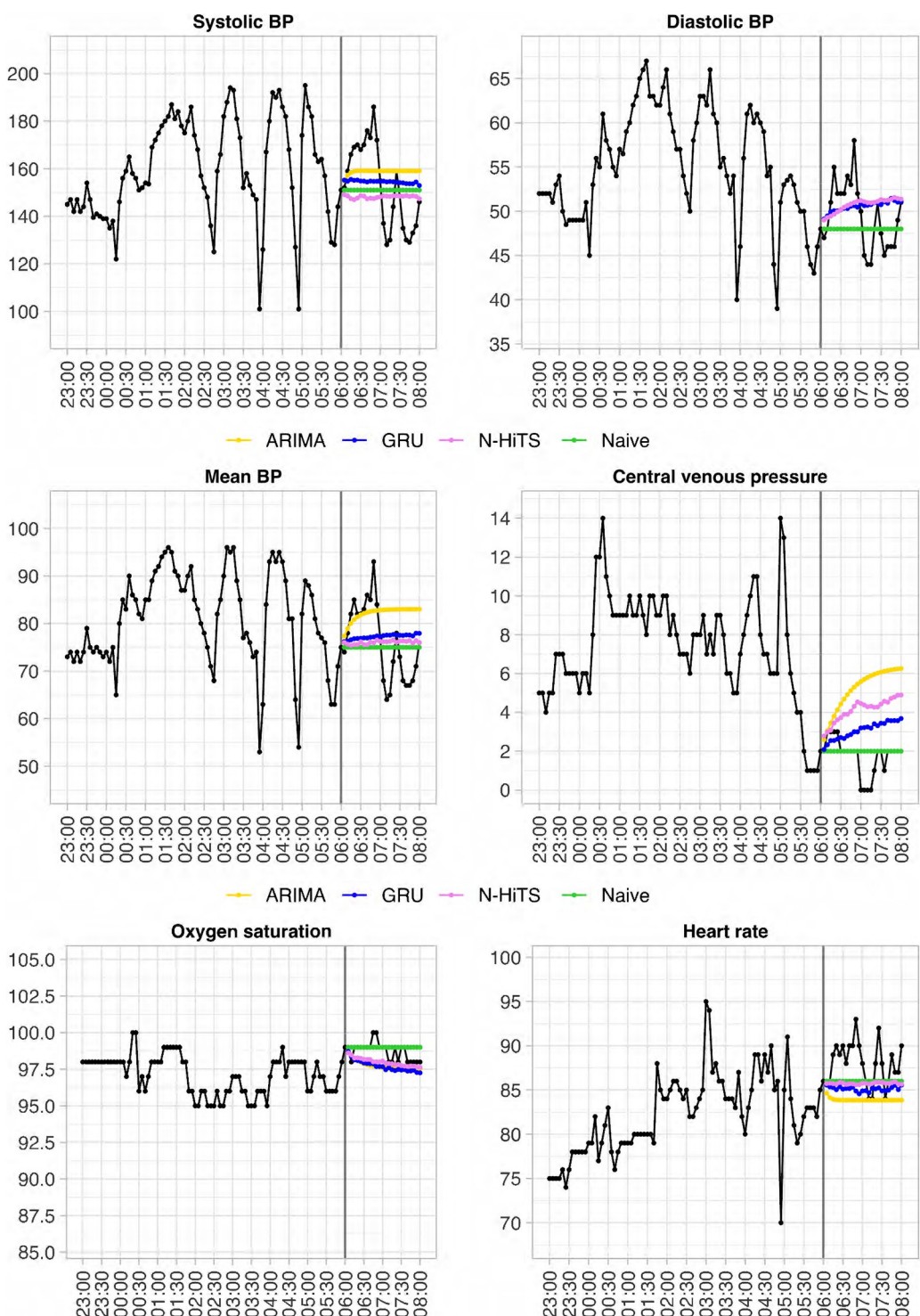

**Fig 2. An example of forecasts produced by four different models at a single timepoint for a single patient for all six vital parameters.** Times are shifted for additional de-identification of the patient. The ground truth is displayed in black.

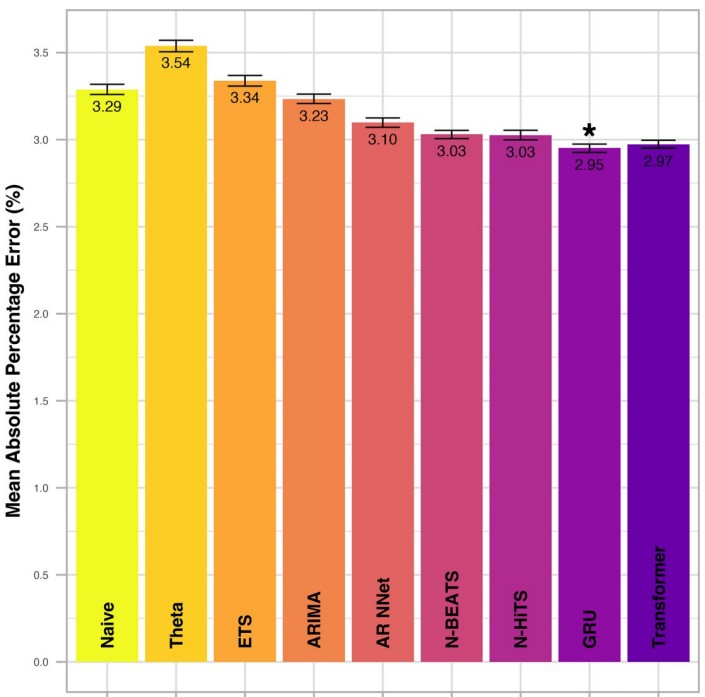

**Fig 3. Mean absolute percentage error (MAPE) across all vital parameters and forecast horizons in the <u>internal</u> <u>test set</u>.** The best performing model (in terms of MAPE) is indicated by *. Error bars indicate the 95% confidence interval and were calculated from 250 bootstrap samples.

For all methods and vital parameters, forecast errors increase with increasing forecasting horizon (Figs 5 and 6). All four implemented neural network models display major improvements over conventional, univariate statistical models. The GRU, however, stands out as the best performing methodology implemented in this study. Its mean absolute percentage error across all vital parameters and forecast horizons is the lowest among all models, both in the internal (Fig 3) and the external test set (Fig 4). The Transformer exhibits the second-best performance both in the internal and external test sets. Thereby, multivariate neural network models claim the top spot among the model classes implemented in this study, followed by univariate neural networks, and lastly univariate statistical models. However, the gap between uni- and multivariate neural network models is not as pronounced as the gap between univariate neural networks and univariate statistical models.

The advantages of the neural network models become more pronounced with increased forecasting horizon (Figs 5 and 6). When forecasting only five minutes into the future, the relative improvement compared to simpler forecasting methods is slight. However, when forecasting 60 or 120 minutes into the future, these methods prove especially beneficial. Further, unlike other methods from the class of univariate statistical models, the four implemented neural network models consistently out-perform the naïve method.

Within the class of univariate neural networks, the difference between N-HiTS and N-BEATS is slight, although N-HiTS exhibits a fractionally smaller MAPE across all vital parameters and forecast horizons both for the internal and the external test sets. Among the univariate statistical models, the AR NNet displays the best forecast accuracy for forecasting horizons larger than 5. However, for forecasts 5 minutes ahead, its performance is sub-par.

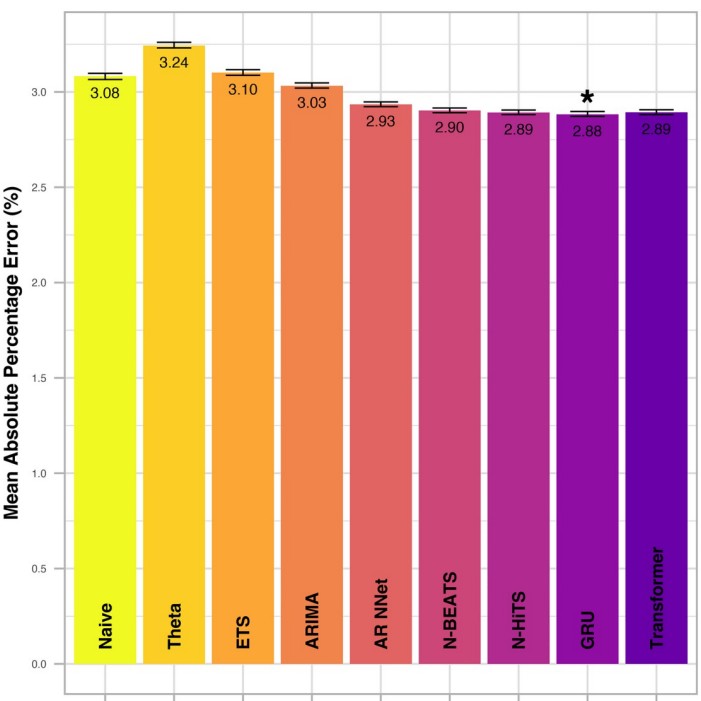

**Fig 4. Mean absolute percentage error (MAPE) across all vital parameters and forecast horizons in the external test set (eICU).** The best performing model (in terms of MAPE) is indicated by *. Error bars indicate the 95% confidence interval and were calculated from 250 bootstrap samples.

Like GRU, Transformer, N-HiTS and N-BEATS, AR NNet's performance relative to the naïve model consistently improves with increasing forecast horizon.

ARIMA performs second best among univariate statistical models, but the improvement upon the naïve model is mostly negligible.

ETS and Theta perform poorly. For large forecast horizons of 60 or 120 minutes, they are consistently the worst performing models, and drastically underperform even when compared to the naïve model (Figs 5 and 6).

## Discussion

In this study, we applied forecasting methodologies of varying complexity to vital parameters tracked in the ICU following cardiothoracic surgery and compared the achieved forecast accuracies. Data originated from roughly 32,000 episodes of ICU care from a German heart center and the American multicenter eICU collaborative research database. With six vital parameters per patient, nine forecasting models, and forecast horizons ranging from 5 to 120 minutes, we established the most extensive set of benchmarks of forecast performance in the ICU to date.

We found modern neural network techniques to be far superior to conventional statistical models, with a widening performance gap for higher forecast horizons. While forecasting is a statistical learning problem where additional model complexity is oftentimes not rewarded with improved forecast accuracy [13,14], our findings support the initial hypothesis that the ICU is a setting primed for successful application of neural networks for forecasting: Despite high volatility and unpredictability on an individual patient level, there are common elements between patients and their vital parameter curves that can be learned and exploited. Firstly,

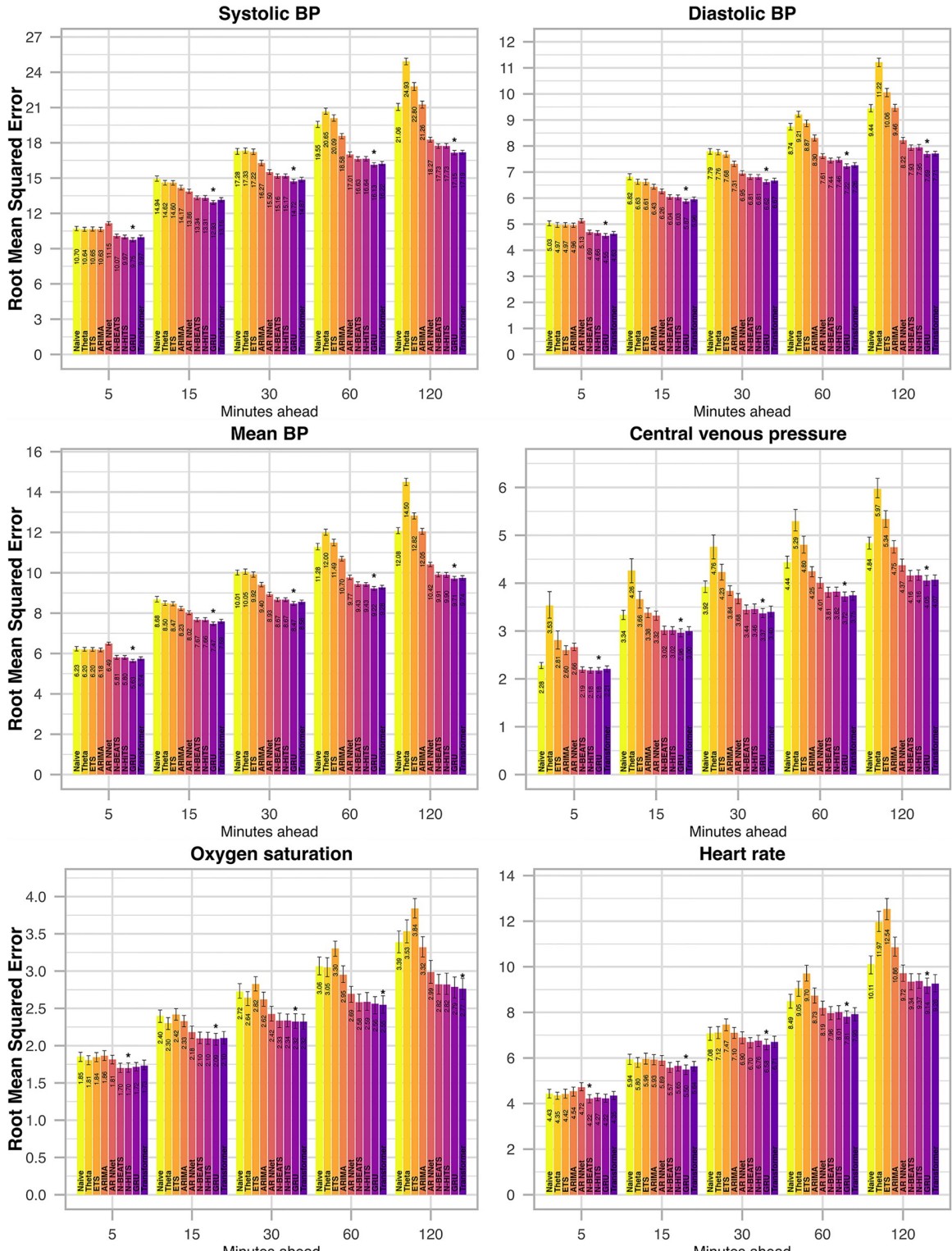

**Fig 5. Root mean squared error (RMSE) per vital parameter for selected forecast horizons in the <u>internal test set</u>.** Different forecasting models are indicated by color. The best performing model (in terms of RMSE) is indicated by *. Error bars indicate the 95% confidence interval and were calculated from 250 bootstrap samples.

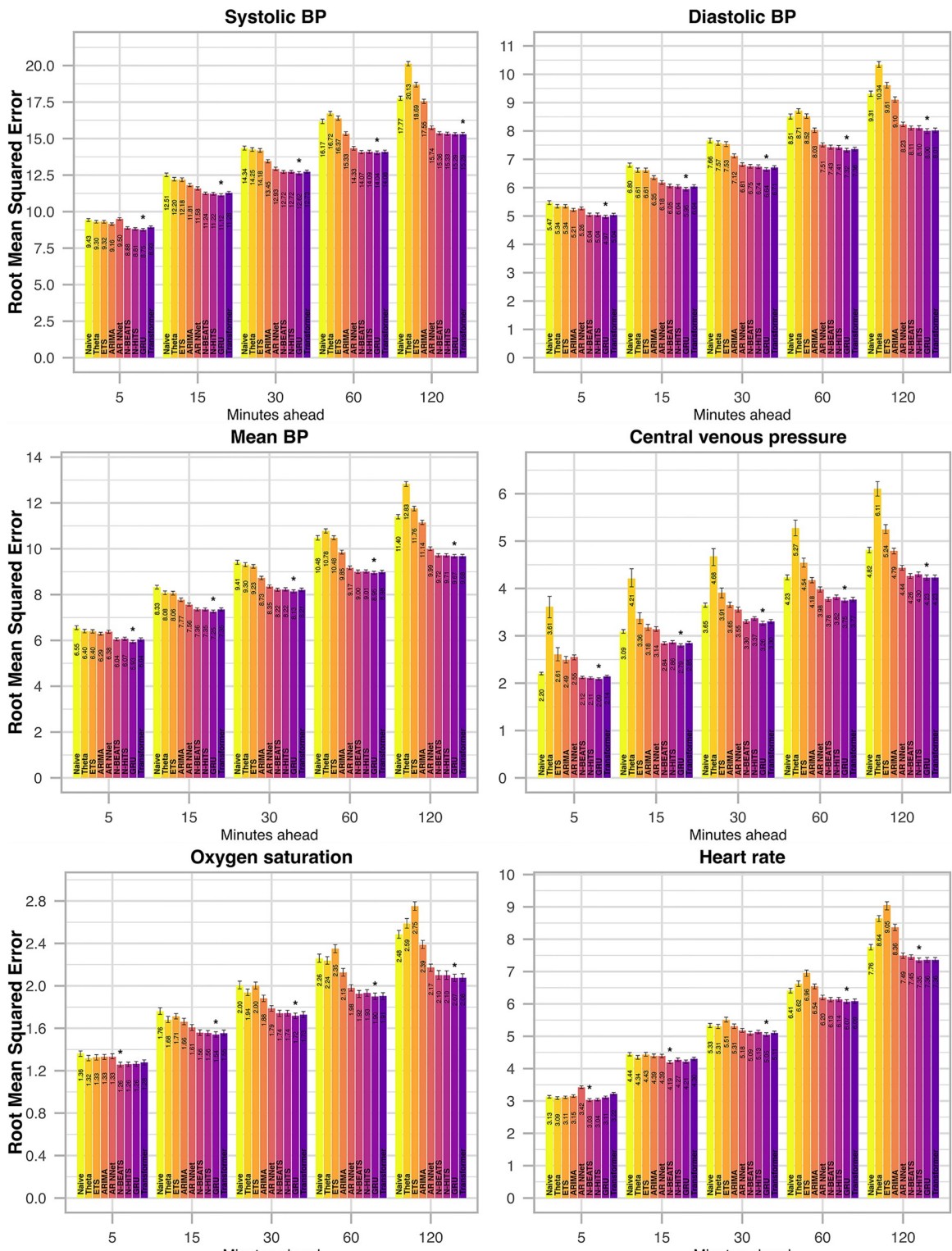

**Fig 6. Root mean squared error (RMSE) per vital parameter for selected forecast horizons in the external test set (eICU).** Different forecasting models are indicated by color. The best performing model (in terms of RMSE) is indicated by *. Error bars indicate the 95% confidence interval and were calculated from 250 bootstrap samples.

certain dynamics, correlations, and limits of human physiology apply to all patients and can be approximated by a model. Secondly, treatment decisions by the medical staff adhere to rigorous guidelines, with the goal of achieving or maintaining vital parameter targets conductive to patient recovery. For example, for most patients lower end target values for mean blood pressure and oxygen saturation are 65 mmHg [32] and 92% [33], respectively. By training multivariate neural networks on close to 20,000 ICU admissions, our neural network models were given ample chance to detect, and at prediction time reflect these patterns. Our findings could therefore be seen as further evidence for the advantages of simultaneously learning from many independent, yet related examples (cross-learning) in time-series forecasting [34].

Their inability to perform cross-patient learning might also be at the core of the inability of some of the implemented univariate statistical models to produce better results than the naïve method. An over-reliance on recent trends could explain poor forecast performance, especially for longer forecast horizons, of the Theta method and ETS. Since these models are fitted independently for each patient, they have no way of judging whether certain trends and patterns ever occur in the ICU, where patients are under constant surveillance with the ability to intervene at short notice in case of unwanted developments.

Regarding the superiority of multivariate over univariate neural network models, we argue that modeling all vital parameters at once diminishes negative effects of missing values or artefacts not detected in the initial data cleaning step, so long as they do not apply to all vital parameters at the same time. Multivariate modeling further adds the potential to draw conclusions on temporal dynamics across vital parameters, e.g. if a worsening condition manifests itself in some vital parameters faster than in others.

Given the recent success of Transformer architectures in tasks requiring the processing of sequential data [20,35], our findings regarding the superiority of the GRU to the Transformer in ICU forecasting is somewhat surprising. While the Transformer is competitive and performs better than univariate models in ICU forecasting, it has not proven to be the state-of-the-art in this specific application. Whether a different choice of Transformer architecture could remedy this shortcoming, or whether we have found the sweet spot of model complexity with the GRU remains speculative.

Regarding the data source used in this study for external validation, we see great potential for the eICU collaborative research database to become one of the benchmark datasets for multivariate time-series forecasting. Due to its large number of ICU stays, multi-center structure, and accessibility for research, evaluation on the eICU database is a formidable test of a model's ability to generalize well and robustly.

## Limitations

Our work exhibits five key limitations, which should be addressed in future work. First, our forecasts were based solely on the vital parameters in question. However, for most patients in intensive care, the health status is captured through additional data sources which we left untapped in this study. Blood tests, medication, diagnoses, information on the preceding surgery or mechanical circulatory support might contain valuable information which could be exploited to further improve forecast accuracies. As currently constructed, the neural networks trained on thousands of ICU stays might have indirectly learned to price in interventions by the medical staff. Adding medication as external features might decrease the risk of models falsely predicting trends that are in truth the result of pharmacological intervention. We refrain from adding external features at this point because a) they are not needed to establish benchmarks or prove the superiority of modern neural network techniques in ICU forecasting, which we were able to demonstrate even without external features, b) their addition comes

with considerable data processing work, especially in multicenter datasets that use non-standardized terms and units of measurement, and c) the process of distinguishing high-impact features that improve forecast performance from those that merely add complexity without additional useable information constitutes a separate research project.

Second, our models produced only point forecasts, whereas in practice prediction intervals might be required to indicate the uncertainty associated with each forecast. While obtaining prediction intervals from statistical models like ARIMA and exponential smoothing is straightforward under strong assumptions and implemented in frameworks like R's forecast package [24], it is a much more complex endeavor for recurrent neural networks or Transformers, and requires the implementation of Bayesian modeling, quantile regression, repeated sampling techniques or conformal predictions [36,37]. Given the diverse approaches to this problem and the difficulty of evaluating intervals due to their inherent tradeoff between width and coverage rate [38], the derivation of prediction intervals is an ongoing research topic that deserves dedicated exploration.

Third, we developed and validated our models on a highly specific patient cohort, postoperative cardiothoracic surgery patients. Whether our findings translate to other ICU cohorts, and whether specific cohorts require specific forecast models remains to be tested.

Fourth, we evaluated our models strictly on quantitative performance in terms of deviation from the ground truth, regardless of medical context or interpretation of predicted changes. How specific medical scenarios influence forecasting, and in which cases it could be most beneficially employed is beyond the scope of this benchmark study.

Fifth and finally, this study does not address practical questions regarding optimal visualization and operationalization in real world ICU settings. On the path towards forecast-guided decision support in the ICU, details regarding the optimal frequency of forecasts, the most beneficial forecast horizon and the visual or acoustical highlighting of forecasts exceeding specified limits need to be customized to different medical domains and ICU workflows.

## Conclusions

Generating accurate short-term forecasts of vital parameters for postoperative patients in intensive care is feasible, and displaying forecasts to the medical staff offers the potential to improve patient care through foresighted decision-making. Whilst therapeutic interventions will never be solely based on model-based forecasts, we see the potential for further development into a decision support tool for automated early warning to trigger a review by the patient's caregiver.

Regarding the choice of forecast models, modern multivariate neural networks achieve forecast accuracies superior to univariate time series models. Due to their demonstrated advantage in terms of accuracy, and their potential for further improvements through incorporating external parameters such as administered medication or laboratory results, they should be a focus for the further development of forecasting techniques in the ICU.

## Supporting information

**S1 Fig. Comparison of distributions of vital parameters between internal (training, validation, test) cohorts and the external test cohort (eICU).**
(TIF)

**S1 Table. Fraction of missing values per vital parameter.** For the internal dataset, missingness is reported pre and post resampling from 1- to 5-minute frequency.
(DOCX)

**S2 Table. Root mean squared error (RMSE), mean absolute error (MAE), and mean absolute percentage error (MAPE) per vital parameter, forecast model, forecast horizon and dataset (internal and external test sets).** BP = Blood pressure. SpO2 = Peripheral oxygen saturation. ETS = Exponential smoothing. ARIMA = Autoregressive integrated moving average. AR NNet = Autoregressive neural network. GRU = Gated recurrent unit.
(DOCX)

**S3 Table. Optimal hyperparameters of the Transformer model.**
(DOCX)

**S4 Table. Optimal hyperparameters of the GRU model.**
(DOCX)

**S5 Table. Optimal hyperparameters of the N-BEATS model per vital parameter.**
(DOCX)

**S6 Table. Optimal hyperparameters of the N-HiTS model per vital parameter.**
(DOCX)

## Acknowledgments

Tobias Roeschl is participant in the BIH Charité Digital Clinician Scientist Program funded by the Charité–Universitätsmedizin Berlin, and the Berlin Institute of Health at Charité (BIH).

## Author Contributions

**Conceptualization:** Nils Hinrichs, Carsten Eickhoff, Alexander Meyer.

**Data curation:** Nils Hinrichs, Tobias Roeschl.

**Formal analysis:** Nils Hinrichs, Carsten Eickhoff, Alexander Meyer.

**Funding acquisition:** Volkmar Falk, Alexander Meyer.

**Investigation:** Nils Hinrichs, Tobias Roeschl, Pia Lanmueller, Carsten Eickhoff, Alexander Meyer.

**Methodology:** Nils Hinrichs, Alexander Meyer.

**Project administration:** Alexander Meyer.

**Resources:** Volkmar Falk, Alexander Meyer.

**Software:** Nils Hinrichs.

**Supervision:** Alexander Meyer.

**Validation:** Nils Hinrichs, Tobias Roeschl, Carsten Eickhoff, Alexander Meyer.

**Visualization:** Nils Hinrichs.

**Writing – original draft:** Nils Hinrichs, Tobias Roeschl.

**Writing – review & editing:** Nils Hinrichs, Tobias Roeschl, Pia Lanmueller, Felix Balzer, Carsten Eickhoff, Benjamin O'Brien, Volkmar Falk, Alexander Meyer.

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
