## [Decision Letter · Decision Letter 0]

27 Feb 2024

PDIG-D-23-00376

Short-term vital parameter forecasting in the intensive care unit - A benchmark study leveraging data from patients after cardiothoracic surgery

PLOS Digital Health

Dear Dr. Hinrichs,

Thank you for submitting your manuscript to PLOS Digital Health. After careful consideration, we feel that it has merit but does not fully meet PLOS Digital Health's publication criteria as it currently stands. Therefore, we invite you to submit a revised version of the manuscript that addresses the points raised during the review process.

Please submit your revised manuscript within 60 days Apr 27 2024 11:59PM. If you will need more time than this to complete your revisions, please reply to this message or contact the journal office at digitalhealth@plos.org. Please include the following items when submitting your revised manuscript:

We look forward to receiving your revised manuscript.

Kind regards,

Sagar Barage, Ph.D.

Guest Editor

PLOS Digital Health

Journal Requirements:

1. Please send a completed 'Competing Interests' statement, including any COIs declared by your co-authors. If you have no competing interests to declare, please state "The authors have declared that no competing interests exist". Otherwise please declare all competing interests beginning with the statement "I have read the journal's policy and the authors of this manuscript have the following competing interests:"

2. Please provide separate figure files in .tif or .eps format only and remove any figures embedded in your manuscript file. Please also ensure that all files are under our size limit of 10MB.

Additional Editor Comments (if provided):

Dear Author,

The important question has been addressed in the manuscript. However, the major concern related to dataset and models utilized by author for prediction. We encourage author to revise and resubmit the manuscript based on the comments raised by reviewer. 

Reviwer 1#

General

The paper addresses important questions and is also methodologically sound, as the results are also successfully validated externally. It is also beneficial that the code has been published online.

In general, the quality of English could be improved. Some sentences are not easy to understand, and some technical terms could be replaced by more easily understandable words for the general medical audience. 

Most important issue: All models do not really seem to react to the particular details of the vital parameters and only approximate the statistical average value of the respective vital parameter based on the current value. Only the vital sign data was used to predict the vital signs, thus the model cannot account for any medication, patient characteristics or treatment influences, which all have a huge effect on the vital signs. This is probably also the reason why the model is not able to show more complex courses. 

5 vital parameter forecasting in the ICU, it therefore remains to be established whether an increase in model

complexity yields better forecasting performance.

The authors do not seem to use the optimal forecasting models: See: https://nixtlaverse.nixtla.io/neuralforecast/index.html

For compatibility of the two data sources, we down-sampled vital parameters in the internal single-center cohort to five-minute intervals using the median of the spanned values.

This has the effect of making the vital parameters extremely smooth: A rapid change is particularly interesting for a parameter such as oxygen saturation, which is not recorded in this way.

Have values from the future been used for this averaging? Was the interval [0.5] or [2.5, 7.5] used for the averaging? This could potentially provide the algorithm with information from the future.

Thus, each ICU admission corresponds to one unique patient and vice versa.

Have you made sure to keep readmissions in only one data set? A patient who is in the data set several times should not appear in the test and the training set. Otherwise, the accuracy is fictitiously increased.

Using pre-defined bounds of plausibility for each vital parameter (Table 1), we removed values exceeding

these limits and considered them missing.

Was this done before or after smoothing? The boundaries seem very narrow. Especially with vital parameters, you should not delete outliers, as these sometimes have a particularly high value. What about patients who have died in the ICU? Their vital signs should be below the lower limits.

Forecasts were always based on the entire observed past up to the current time point. Thus, the length of the forecast window remained fixed at 120 minutes, but the amount of past values on which predictions were based depended on the time point of prediction

How was this achieved? It is not trivial that a model can use different input lengths. Was it shortened or was the input padded? Does the model work from the first minute or only from a certain point in time?

Figure 1: Schematic illustration of the forecasting setup. Synthetic example of three successive forecasts produced by the same method for one vital parameter of a single patient. Layout of the figure inspired by [11].

The model appears to output the average value of the vital parameter but does not react to the individual change of the patient.

In contrast, for the multivariate neural networks, we trained a single, immutable model on all patients and vital parameters observed in the training set

Have you tried to train the neuronal networks for just one vital sign?

Per patient and vital parameter, we produced forecasts every five minutes starting with the sixth, and ending

with the penultimate timepoint. Thus, univariate models were given at least five past observations to train on,

and all forecasts could be evaluated on at least one true value.

Does the model need 25 (5x5) minutes of history to work?

If at least 24 past observations were available for a given vital parameter, we used Box-Cox

transformation[22] on the time series with a transformation parameter between zero and two, using the

implementation in the forecast package[21].

What was done before 24 observations? If the other input parameters are scaled, the first 24 must also be scaled.

How was the output scaled back?

MAE, MSME are dependent on the scaling of the data set, therefore this scaling is of great importance, otherwise it is difficult to compare the deviations with other publications. Which I think is an important point of this publication.

Vital parameters were standardized to exhibit mean zero and standard deviation one prior to

model fitting, and de-standardized post prediction.

Which data set was used to create the standard scaler? Technically, only the training set should be used for this and the other sets should also be scaled with the scalers from the training set.

The models were designed to output forecasts for all 24 forecast horizons (5 through 120 minutes) simultaneously.

How was the variable length of the input time horizon considered?

We used the HyperOpt search algorithm[26] with trials scheduled and ended prematurely in case of poor performance after at least 10 epochs using Hyperband scheduling[27]. Models were trained for up to 100 epochs with training data presented in batches of 128 patients. 50 different hyperparameter combinations were tried to find the most suitable model

Which methods did the authors use to minimise overfitting?

Neural network model parameters For the GRU model, hyperparameter tuning resulted in a single-layer model with hidden size 256, a dropout of 0.37, L2 regularization parameter of 6 · 10-6, which was fitted using a learning rate of 1.31 · 10-3. The optimal Transformer had 3 layers, model dimension 128, hidden size 256, 16 transformer heads, used fixed positional encoding, the rectified linear unit as activation function, a dropout of 0.04, L2 regularization

parameter 4.45 · 10-5, and was fitted using a learning rate of 1.25 · 10-3.

This Information could be transferred to the supplement section.

Out of the 144 combinations of vital parameter and forecast horizon, the GRU exhibits the smallest RMSE 136

imes (94.44%) in the internal test set, and 119 times (82.64%) in the external test set.

I think an overall MAP and a figure showing this information would be preferable.

Figure 2: An example of forecasts produced by four different models at a single timepoint for a single patient for all six

vital parameters. Times are shifted for additional de-identification of the patient. The ground truth is displayed in black.

The model does not seem to be able to predict the detailed change in vital signs, but only the trend value. What then is the advantage over a model that simply predicts the value for 2 hours in the future and then interpolates this value?

Figure 3: Root mean squared error (RMSE) per vital parameter for selected forecast horizons in the internal test set.

Different forecasting models are indicated by color. The best performing model (in terms of RMSE) is indicated by *.

Error bars would be advantageous. What does ‘selected forecast horizons’ mean? Why wasn't the average of all patients in the test set taken?

Figure 5: Relative improvement on the root mean squared error (RMSE) of the naïve model per vital parameter for

selected forecast horizons in the internal test set. Different forecasting models are indicated by color. The best performing model (in terms of RMSE) is indicated by *.

Personally, I don't find the information in this graphic essential.

For example, for most patients lower end target values for mean blood pressure and oxygen saturation are 65 mmHg[28] and 92%[29], respectively. By training multivariate neural networks on close to 20,000 ICU admissions, our models were given ample chance to detect, and at prediction time reflect these patterns.

It would be very interesting to see if the model can predict lower end target values for mean blood pressure and oxygen saturation.

Given the recent success of Transformer architectures in tasks requiring the processing of sequential data[17,

31], our findings regarding the superiority of the GRU to the Transformer in ICU forecasting is somewhat

surprising. While the Transformer is competitive and performs better than univariate statistical models in ICU

Why have you not tested any other Transformer models?

Our work exhibits four key limitations, which should be addressed in future work. First, our forecasts were

based solely on the vital parameters in question.

Have you tried using more input parameters? This should be stated more clearly in the methodology. 

While obtaining confidence intervals from statistical models like ARIMA and exponential smoothing is straightforward and implemented in frameworks like R’s forecast package[21], it is a much more complex endeavor for recurrent neural networks or Transformers, and requires the implementation of Bayesian modeling, quantile regression or repeated sampling techniques[32].

It is more complex, but very doable. You can also calculate a confidence interval from the number of errors.

Reviwer 2#

Thank you for submitting this interesting research.

The study focuses on short-term forecasting of vital parameters in ICU patients post-cardiothoracic surgery. It compares various univariate statistical models and multivariate neural networks for predicting vital signs. The research finds multivariate neural networks, particularly GRU and Transformer models, outperform univariate models especially for longer forecast horizons. The study establishes comprehensive benchmarks for ICU vital parameter forecasting, suggesting the feasibility and utility of AI-driven models in clinical decision-making in ICU settings.

However, I believe this study contains the following issues to be considered by authors:

The major practical concern is that the predicted parameters are not easily utilized in real clinical setting - which actually should be the biggest limitation of all (to be in the limitation section) for this work. The evaluation of the prediction performance of vital signs by 4 different models suffer from the lack of visualization. For example, the Figure 2 shows a very long horizon of prediction of each model at the time of prediction - and each color flies out only based on the fixed time. The authors could show the temporal model performance by showing the model to be on the moving windows - hence visualizing the model accuracy (MAPE or MAE + standard deviation) in regression metrics better, as the time goes to the future. For readers, any of those 4 models seem to be suboptimally performing. Authors might want to present quantitative difference between the model and real performances - supplying Bland-Altman plots could help as well. Also it would be beneficial for the paper to include the techniques and approaches used to model these dynamic changes, as well as an analysis of how these temporal variations impact the prediction outcomes.

Some minor comments as well as below:

Line 98 – 99: Has the study accounted for the potential noise in the data within the first 30 mins to 1 hour after ICU admission? This initial period can often include transitional dynamics that may not be representative of the patients’ stable physiological state.

Line 109 – 111: Could the authors provide the percentage or extent of missing data encountered in this study? Understanding the volume of missing values is crucial for assessing the robustness of the imputation techniques and the reliability of the models’ predictions, especially in a complex ICU setting. Further, both of ‘Forward-filling’ or ‘Median imputation’ might not account for individual patient variability and trends, which are critical in the ICU setting. This could affect the models’ ability to accurately predict outcomes. More sophisticated techniques like time-series imputation methods or predictive modeling could potentially provide more accurate ways to handle missing data.

Line 200 – 206: In the comparison between the internal cohorts and the external test set, it is observed that different statistical measures are used: for instance, the lower quartile for central venous pressure and the median for heart rates. It would be helpful to know if these differences were statistically significant.

Reviewers' comments:

Reviewer's Responses to Questions

**Comments to the Author**

1. Does this manuscript meet PLOS Digital Health’s publication criteria? Is the manuscript technically sound, and do the data support the conclusions? The manuscript must describe methodologically and ethically rigorous research with conclusions that are appropriately drawn based on the data presented.

Reviewer #1: Yes

Reviewer #2: Yes

2. Has the statistical analysis been performed appropriately and rigorously?

Reviewer #1: Yes

Reviewer #2: No

3. Have the authors made all data underlying the findings in their manuscript fully available (please refer to the Data Availability Statement at the start of the manuscript PDF file)?

Reviewer #1: Yes

Reviewer #2: Yes

4. Is the manuscript presented in an intelligible fashion and written in standard English?

Reviewer #1: Yes

Reviewer #2: Yes

5. Review Comments to the Author

Reviewer #1: also see attachment with original text in bold letters. 

5 vital parameter forecasting in the ICU, it therefore remains to be established whether an increase in model

complexity yields better forecasting performance.

The authors do not seem to use the optimal forecasting models: See: https://nixtlaverse.nixtla.io/neuralforecast/index.html

For compatibility of the two data sources, we down-sampled vital parameters in the internal single-center cohort to five-minute intervals using the median of the spanned values.

This has the effect of making the vital parameters extremely smooth: A rapid change is particularly interesting for a parameter such as oxygen saturation, which is not recorded in this way.

Have values from the future been used for this averaging? Was the interval [0.5] or [2.5, 7.5] used for the averaging? This could potentially provide the algorithm with information from the future.

Thus, each ICU admission corresponds to one unique patient and vice versa.

Have you made sure to keep readmissions in only one data set? A patient who is in the data set several times should not appear in the test and the training set. Otherwise, the accuracy is fictitiously increased.

Using pre-defined bounds of plausibility for each vital parameter (Table 1), we removed values exceeding

these limits and considered them missing.

Was this done before or after smoothing? The boundaries seem very narrow. Especially with vital parameters, you should not delete outliers, as these sometimes have a particularly high value. What about patients who have died in the ICU? Their vital signs should be below the lower limits.

Forecasts were always based on the entire observed past up to the current time point. Thus, the length of the forecast window remained fixed at 120 minutes, but the amount of past values on which predictions were based depended on the time point of prediction

How was this achieved? It is not trivial that a model can use different input lengths. Was it shortened or was the input padded? Does the model work from the first minute or only from a certain point in time?

Figure 1: Schematic illustration of the forecasting setup. Synthetic example of three successive forecasts produced by the same method for one vital parameter of a single patient. Layout of the figure inspired by [11].

The model appears to output the average value of the vital parameter but does not react to the individual change of the patient.

In contrast, for the multivariate neural networks, we trained a single, immutable model on all patients and vital parameters observed in the training set

Have you tried to train the neuronal networks for just one vital sign?

Per patient and vital parameter, we produced forecasts every five minutes starting with the sixth, and ending

with the penultimate timepoint. Thus, univariate models were given at least five past observations to train on,

and all forecasts could be evaluated on at least one true value.

Does the model need 25 (5x5) minutes of history to work?

If at least 24 past observations were available for a given vital parameter, we used Box-Cox

transformation[22] on the time series with a transformation parameter between zero and two, using the

implementation in the forecast package[21].

What was done before 24 observations? If the other input parameters are scaled, the first 24 must also be scaled.

How was the output scaled back?

MAE, MSME are dependent on the scaling of the data set, therefore this scaling is of great importance, otherwise it is difficult to compare the deviations with other publications. Which I think is an important point of this publication.

Vital parameters were standardized to exhibit mean zero and standard deviation one prior to

model fitting, and de-standardized post prediction.

Which data set was used to create the standard scaler? Technically, only the training set should be used for this and the other sets should also be scaled with the scalers from the training set.

The models were designed to output forecasts for all 24 forecast horizons (5 through 120 minutes) simultaneously.

How was the variable length of the input time horizon considered?

We used the HyperOpt search algorithm[26] with trials scheduled and ended prematurely in case of poor performance after at least 10 epochs using Hyperband scheduling[27]. Models were trained for up to 100 epochs with training data presented in batches of 128 patients. 50 different hyperparameter combinations were tried to find the most suitable model

Which methods did the authors use to minimise overfitting?

Neural network model parameters For the GRU model, hyperparameter tuning resulted in a single-layer model with hidden size 256, a dropout of 0.37, L2 regularization parameter of 6 · 10-6, which was fitted using a learning rate of 1.31 · 10-3. The optimal Transformer had 3 layers, model dimension 128, hidden size 256, 16 transformer heads, used fixed positional encoding, the rectified linear unit as activation function, a dropout of 0.04, L2 regularization

parameter 4.45 · 10-5, and was fitted using a learning rate of 1.25 · 10-3.

This Information could be transferred to the supplement section.

Out of the 144 combinations of vital parameter and forecast horizon, the GRU exhibits the smallest RMSE 136

imes (94.44%) in the internal test set, and 119 times (82.64%) in the external test set.

I think an overall MAP and a figure showing this information would be preferable.

Figure 2: An example of forecasts produced by four different models at a single timepoint for a single patient for all six

vital parameters. Times are shifted for additional de-identification of the patient. The ground truth is displayed in black.

The model does not seem to be able to predict the detailed change in vital signs, but only the trend value. What then is the advantage over a model that simply predicts the value for 2 hours in the future and then interpolates this value?

Figure 3: Root mean squared error (RMSE) per vital parameter for selected forecast horizons in the internal test set.

Different forecasting models are indicated by color. The best performing model (in terms of RMSE) is indicated by *.

Error bars would be advantageous. What does ‘selected forecast horizons’ mean? Why wasn't the average of all patients in the test set taken?

Figure 5: Relative improvement on the root mean squared error (RMSE) of the naïve model per vital parameter for

selected forecast horizons in the internal test set. Different forecasting models are indicated by color. The best performing model (in terms of RMSE) is indicated by *.

Personally, I don't find the information in this graphic essential.

For example, for most patients lower end target values for mean blood pressure and oxygen saturation are 65 mmHg[28] and 92%[29], respectively. By training multivariate neural networks on close to 20,000 ICU admissions, our models were given ample chance to detect, and at prediction time reflect these patterns.

It would be very interesting to see if the model can predict lower end target values for mean blood pressure and oxygen saturation.

Given the recent success of Transformer architectures in tasks requiring the processing of sequential data[17,

31], our findings regarding the superiority of the GRU to the Transformer in ICU forecasting is somewhat

surprising. While the Transformer is competitive and performs better than univariate statistical models in ICU

Why have you not tested any other Transformer models?

Our work exhibits four key limitations, which should be addressed in future work. First, our forecasts were

based solely on the vital parameters in question.

Have you tried using more input parameters? This should be stated more clearly in the methodology. 

While obtaining confidence intervals from statistical models like ARIMA and exponential smoothing is straightforward and implemented in frameworks like R’s forecast package[21], it is a much more complex endeavor for recurrent neural networks or Transformers, and requires the implementation of Bayesian modeling, quantile regression or repeated sampling techniques[32].

It is more complex, but very doable. You can also calculate a confidence interval from the number of errors.

General

The paper addresses important questions and is also methodologically sound, as the results are also successfully validated externally. It is also beneficial that the code has been published online.

In general, the quality of English could be improved. Some sentences are not easy to understand, and some technical terms could be replaced by more easily understandable words for the general medical audience. 

Most important issue: All models do not really seem to react to the particular details of the vital parameters and only approximate the statistical average value of the respective vital parameter based on the current value. Only the vital sign data was used to predict the vital signs, thus the model cannot account for any medication, patient characteristics or treatment influences, which all have a huge effect on the vital signs. This is probably also the reason why the model is not able to show more complex courses. Also the preprocessing contains a few inaccuracies (see above).

Reviewer #2: Thank you for submitting this interesting research. 

The study focuses on short-term forecasting of vital parameters in ICU patients post-cardiothoracic surgery. It compares various univariate statistical models and multivariate neural networks for predicting vital signs. The research finds multivariate neural networks, particularly GRU and Transformer models, outperform univariate models especially for longer forecast horizons. The study establishes comprehensive benchmarks for ICU vital parameter forecasting, suggesting the feasibility and utility of AI-driven models in clinical decision-making in ICU settings.

However, I believe this study contains the following issues to be considered by authors:

The major practical concern is that the predicted parameters are not easily utilized in real clinical setting - which actually should be the biggest limitation of all (to be in the limitation section) for this work. The evaluation of the prediction performance of vital signs by 4 different models suffer from the lack of visualization. For example, the Figure 2 shows a very long horizon of prediction of each model at the time of prediction - and each color flies out only based on the fixed time. The authors could show the temporal model performance by showing the model to be on the moving windows - hence visualizing the model accuracy (MAPE or MAE + standard deviation) in regression metrics better, as the time goes to the future. For readers, any of those 4 models seem to be suboptimally performing. Authors might want to present quantitative difference between the model and real performances - supplying Bland-Altman plots could help as well. Also it would be beneficial for the paper to include the techniques and approaches used to model these dynamic changes, as well as an analysis of how these temporal variations impact the prediction outcomes.

Some minor comments as well as below:

Line 98 – 99: Has the study accounted for the potential noise in the data within the first 30 mins to 1 hour after ICU admission? This initial period can often include transitional dynamics that may not be representative of the patients’ stable physiological state.

Line 109 – 111: Could the authors provide the percentage or extent of missing data encountered in this study? Understanding the volume of missing values is crucial for assessing the robustness of the imputation techniques and the reliability of the models’ predictions, especially in a complex ICU setting. Further, both of ‘Forward-filling’ or ‘Median imputation’ might not account for individual patient variability and trends, which are critical in the ICU setting. This could affect the models’ ability to accurately predict outcomes. More sophisticated techniques like time-series imputation methods or predictive modeling could potentially provide more accurate ways to handle missing data.

Line 200 – 206: In the comparison between the internal cohorts and the external test set, it is observed that different statistical measures are used: for instance, the lower quartile for central venous pressure and the median for heart rates. It would be helpful to know if these differences were statistically significant.

6. PLOS authors have the option to publish the peer review history of their article (what does this mean?). If published, this will include your full peer review and any attached files.

**Do you want your identity to be public for this peer review?** For information about this choice, including consent withdrawal, please see our Privacy Policy.

Reviewer #1: No

Reviewer #2: No

---

## [Decision Letter · Decision Letter 1]

21 Jun 2024

PDIG-D-23-00376R1

Short-term vital parameter forecasting in the intensive care unit - A benchmark study leveraging data from patients after cardiothoracic surgery

PLOS Digital Health

Dear Dr. Hinrichs,

Thank you for submitting your manuscript to PLOS Digital Health. After careful consideration, we feel that it has merit but does not fully meet PLOS Digital Health's publication criteria as it currently stands. Therefore, we invite you to submit a revised version of the manuscript that addresses the points raised during the review process.

Please submit your revised manuscript within 30 days Jul 21 2024 11:59PM. If you will need more time than this to complete your revisions, please reply to this message or contact the journal office at digitalhealth@plos.org. Please include the following items when submitting your revised manuscript:

We look forward to receiving your revised manuscript.

Kind regards,

Sagar Barage, Ph.D.

Guest Editor

PLOS Digital Health

Journal Requirements:

Additional Editor Comments (if provided):

Dear Author,

As most of reviewer queries addressed by author in revised version of manuscript. In general, the algorithms used are simply not that good and tend to predict the average rather than providing good individual results. However, I recommend author to address the reviewer comment and submit the revised version of manuscript for acceptance of the manuscript for publication.

Reviewers' comments:

Reviewer's Responses to Questions

**Comments to the Author**

1. If the authors have adequately addressed your comments raised in a previous round of review and you feel that this manuscript is now acceptable for publication, you may indicate that here to bypass the “Comments to the Author” section, enter your conflict of interest statement in the “Confidential to Editor” section, and submit your "Accept" recommendation.

Reviewer #1: All comments have been addressed

2. Does this manuscript meet PLOS Digital Health’s publication criteria? Is the manuscript technically sound, and do the data support the conclusions? The manuscript must describe methodologically and ethically rigorous research with conclusions that are appropriately drawn based on the data presented.

Reviewer #1: Yes

3. Has the statistical analysis been performed appropriately and rigorously?

Reviewer #1: Yes

4. Have the authors made all data underlying the findings in their manuscript fully available (please refer to the Data Availability Statement at the start of the manuscript PDF file)?

Reviewer #1: Yes

5. Is the manuscript presented in an intelligible fashion and written in standard English?

Reviewer #1: Yes

6. Review Comments to the Author

Reviewer #1: The manuscript has improved - 

comment #26: would be doable in the reviewer's opinion without a necessary extension in a seperate publication, as the authors state. 

Figure resolution is not appropriate (but maybe due to conversion and adequate in the original files?). 

In general the algorithms used are simply not that good and tend to predict the average rather than providing good individual results.

7. PLOS authors have the option to publish the peer review history of their article (what does this mean?). If published, this will include your full peer review and any attached files.

**Do you want your identity to be public for this peer review?** For information about this choice, including consent withdrawal, please see our Privacy Policy. 

Reviewer #1: No

---

## [Editor Report · Decision Letter 2]

30 Jul 2024

Short-term vital parameter forecasting in the intensive care unit - A benchmark study leveraging data from patients after cardiothoracic surgery

PDIG-D-23-00376R2

Dear Mr Hinrichs,

We are pleased to inform you that your manuscript 'Short-term vital parameter forecasting in the intensive care unit - A benchmark study leveraging data from patients after cardiothoracic surgery' has been provisionally accepted for publication in PLOS Digital Health.

Best regards,

Sagar Barage, Ph.D.

Guest Editor

PLOS Digital Health